


# The role of heat wave events on the occurrence and persistence of thermal stratification in the southern North Sea

Wei Chen[1], Joanna Staneva[1], Sebastian Grayek[1], Johannes Schulz-Stellenfleth[1], and Jens Greinert[2]

[1]Institute of Coastal Systems-Analysis and Modelling, Helmholtz-Zentrum Hereon, Max-Planck-Str. 1, 21502 Geesthacht, Germany
[2]GEOMAR Helmholtz Center for Ocean Research Kiel, Wischhofstr. 1-3, 24148 Kiel, Germany

**Correspondence:** Wei Chen (wei.chen@hereon.de)

**Abstract.** Extremes in temperatures not only directly affect the marine environment and ecosystems but also have indirect impacts on hydrodynamics and marine life. The role of heat wave events responsible for the occurrence and persistence of thermal stratification was analysed using a fully coupled hydrodynamic and wave model within the framework of the Geesthacht Coupled cOAstal model SysTem (GCOAST) for the North Sea. The model results were assessed against satellite reprocessed

data and in situ observations from field campaigns and fixed MARNET stations. To quantify the degree of stratification, a potential energy anomaly over the water column was calculated. A linear correlation existed between the air temperatures and the potential energy anomaly in the North Sea excluding the Norwegian Trench and the area south of 54°N latitude. Contrary to the northern part of the North Sea, where the water column is stratified in the warming season each year, the southern North Sea is seasonally stratified in years when a heatwave occurs. The influences of heatwaves on the occurrence of summer stratifications

in the southern North Sea are mainly in the form of two aspects, i.e., a rapid rise in sea surface temperature at the early stage of the heatwave period and a relatively higher water temperature during summer than the multiyear mean. Another factor that enhances the thermal stratification in summer is the memory of the water column to cold spells earlier in the year. Differences between the seasonally stratified northern North Sea and the heatwave-induced stratified southern North Sea were attributed to changes in water depth.

## 1  Introduction

Recently, the increased number of extreme events with the global climate change has attracted more attention from research in regional and global earth systems (IPCC, 2012; Herring et al., 2015). Summer heatwaves are extreme meteorological events that frequently occur (Perkins and Alexander, 2013). Consequently, they cause anomalously warm seawater in discrete periods via local air-sea heat flux exchanges, which are also known as marine heatwaves (MHWs) (Pearce et al., 2011; Hobday et al.,

2016). Other causes of MHWs include ocean heat transport (Rouault et al., 2007; Oliver et al., 2017) or remote forcings (Bond et al., 2015; Hu et al., 2017). MHWs have been identified with a trend of increasing intensification (Oliver et al., 2020). Over the last two decades, record-breaking MHW events have been reported in the Mediterranean Sea (Bensoussan et al., 2010), Tasman Sea (Oliver et al., 2017; Perkins-Kirkpatrick et al., 2019), west Australia (Feng et al., 2013), northeast Pacific (Hu et al., 2017), western South Atlantic (Manta et al., 2018) and East China Sea (Tan and Cai, 2018). The study of MHWs has emerged





as a rapidly growing field of research due to its substantial influence on marine hydrodynamics and ecosystems (Wernberg
et al., 2013, 2016; Oliver et al., 2020). In the North Sea, waters are predicted to be warming fastest relative to global levels and
are highly impacted by the extreme meteorological events (Smale et al., 2019; Hobday and Pecl, 2014); the concequences of
MHWs however, have received much less attention (Wakelin et al., 2021).

The North Sea, which is located on the northwest European passive continental margin, connects to the Baltic Sea in the east
and to the Atlantic through the Norwegian Sea in the north and the English Channel in the west (Figure 1). The water depth is
shallow in the large area of the North Sea, except for the Norwegian channel, which has an average depth of 400 m deep with a
maximum of 750 m in the Skagerrak (Otto et al., 1990). The tidal circulation is dominated by the semidiurnal tides $M_2$ and $S_2$
and interacts with the North Sea bathymetry. Sources of turbulent mixing are the tidal currents at the bottom (Simpson et al.,
1994) and the wind-induced waves at the water surface (Staneva et al., 2017), due to the relatively shallow water depth.

During the summer of 2018, extreme climates with record-breaking temperatures were observed in many countries. The
German Weather Service (DWD) observed weekly temperature anomalies of up to $+3 \sim 6°C$ (Imbery et al., 2018). In addition
to the significant social and environmental impacts across West and North Europe, the summer heatwave imposed large pertur-
bations on natural processes in the North Sea. For example, Borges et al. (2019) found that the dissolved methane concentration
in surface waters along the Belgium coast was three times higher during the summer of 2018 than during a normal year. The
European 2018 heatwave events provide a good opportunity for investigating the impact of extreme climate conditions on the
development of density stratification in the southern North Sea. An identification of extreme temperature events and how these
may have impacts on key fish and shellfish stocks was documented in the recently published Ocean State Report 5 (Wakelin
et al., 2021). However, to our knowledge no systematic studies have yet been performed to explain this impact in relation to
the changes in vertical stratification during extreme temperature events in the North Sea.

As a fundamental physical process, the development of vertical thermal stratification in the North Sea is associated with the
seasonal cycling of water temperatures, especially in the shallow shelf seas. From early May to late September, the majority of
the North Sea becomes stratified, which is also known as the summer stratification (Pingree and Griffiths, 1978). Understanding
the occurrence of summer stratification and its temporal and spatial variations is important and the interest of many studies
(Becker, 1981; Elliott and Clarke, 1991; Pohlmann, 1996; Schrum et al., 2003; Stips et al., 2004; Sharples et al., 2006; van
Leeuwen et al., 2015). In shallow shelf seas, stratification plays an essential role in, e.g., water circulation (van Haren, 2000;
Luyten et al., 2003), sediment flocculation and transport (Fettweis et al., 2014), phytoplankton (Nielsen et al., 1993; Fernand
et al., 2013) and fishery resources (Sas et al., 2019)

Many relevant modelling studies have been conducted within the North Sea (Luyten et al., 2003; Stips et al., 2004; Sharples
et al., 2006; Mathis et al., 2015). More recently, the development of a state-of-the-art three dimensional hydrodynamic model
based on ROMS was demonstrated by Klonaris et al. (2021), who showed accurately reproduced thermohaline variations in
the southern North Sea. Considering the interface between the air and the ocean, models that couple the interactions between
the air-sea system were integrated to investigate thermodynamic air-ocean interface processes (Ho-Hagemann et al., 2017;
Stathopoulos et al., 2020). Staneva et al. (2021) presented a new high-resolution model (part of the Geesthacht COAstal model
SysTem GCOAST) that couples ocean and wave model systems for the North Sea and the Baltic Sea. This model implemented

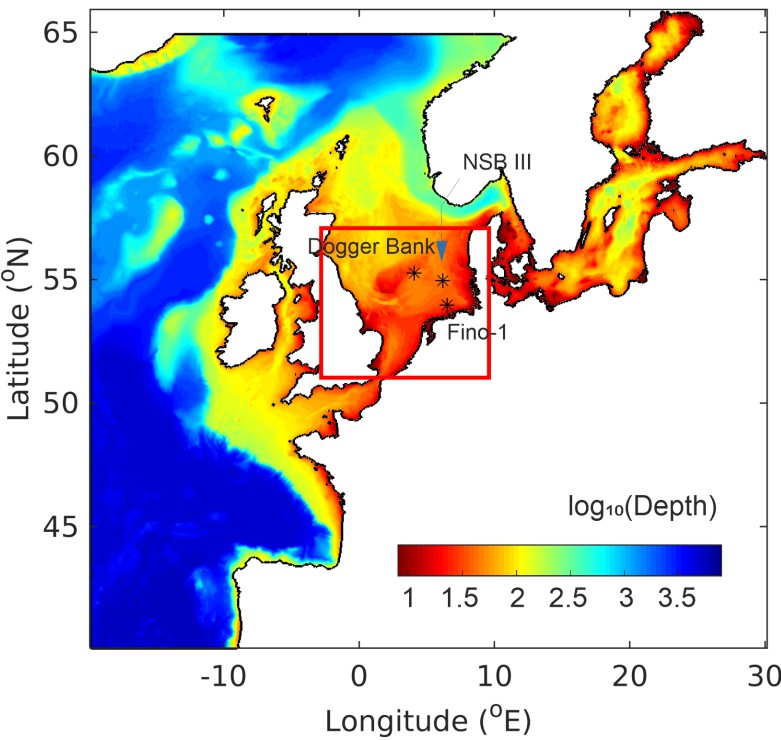

**Figure 1.** The GCOAST model domain. The bathymetry (meters) is shown on a log scale. The location of the observations from the Poseidon cruise (Dogger Bank) and MARNET stations (NSB III and Fino-1) are indicated with asterisks. The red frame defines the area of the middle and southern North Sea.

parameterisations that take into account the nonlinear feedback between tidal currents and wind waves. Within this framework, Chen et al. (2021) developed a 3DVAR data assimilation scheme, which improved sea surface temperature modelling in the North Sea. In that study, the authors further quantified the impact of temperature assimilation on heat budget estimates for the North Sea.

Based on a 51-year simulation data, van Leeuwen et al. (2015) identified five regimes in the North Sea regarding different types of density stratification. The study classified the presence of stratification according to the influence of freshwater from estuaries, permanent haline stratification in the deep Norwegian Trench, seasonal changes in air temperatures, and strong turbulent mixing by tidal currents and waves. However, these classifications fail to categorise approximately 30% of the North Sea area, especially in the southern part, where the water depth is generally less than 50 m and the regime absence of a dominant stratification type accounts for approximately half of the total area. In this area, the density stratification is highly sensitive to the climate conditions and shows interannual variations. The present study focuses on this unclassified area, in order to quantify the role of heatwaves on the occurrence of summer stratification. We further address the question "What are the main factors that affect the intensity and duration of the thermal stratification in the southern North Sea?"



This study applied a fully coupled hydrodynamics and wind-wave model in the North Sea and adjacent seas. The model simulations were compared with recent field measurements of both short-term (a cruise campaign from 23 July to 1 August 2018) and multiyear (6-year stationary measurements). Furthermore, a 38-year sea surface temperature (SST) data reprocessed from satellite observations was used to detect the occurrence and duration of the MHWs. Details of the observations and the methods used for analysis are described in the next section. Section 3 contains the results, and the discussion follows in Section 4. The main conclusions are given in Section 5.

## 2 Materials and Methods

### 2.1 Model

This study conducted an 8-year (2011-2018) numerical simulation with the ocean circulation model NEMO (Madec and the NEMO team, 2016) fully coupled to the wave model WAM (The Wamdi Group, 1988; Günther et al., 1992). The model is set up under the framework of the Geesthacht Coupled cOAstal model SysTem (GCOAST) covering the Northwest European Shelf, the North Sea, and the Baltic Sea, and it has a horizontal resolution of approximately 3.5 km and a vertical resolution of the NEMO standard $\sigma - z^*$ hybrid grid with 50 levels. The present study focuses on the southern North Sea, which is defined as the area between $-3 \sim 9°$E longitude and $51 \sim 57°$N latitude (shown by the red frame in Figure 1).

For boundary conditions, temperature, salinity and barotropic forcing were derived from the hourly Copernicus Marine Environment Monitoring Service (CMEMS) Forecast Ocean Assimilation Model (FOAM) Atlantic Margin Model version 7 (AMM7) output (O'Dea et al., 2012). Tidal harmonic constituents derived from the TPXOv8 model were applied at the open boundaries to force tidal motions (Egbert and Erofeeva., 2002). The ECMWF (European Centre for Medium-Range Weather Forecasts) Reanalysis version 5 (ERA5) data were used at the water surface (Hersbach et al., 2020). The data contained the air temperature 2 m above the water surface, which was also applied for analysing the relation between air temperature and the thermal stratification in the North Sea. The model adopted a baroclinic time step of 100 seconds. Turbulent eddy viscosities/diffusivities were computed with a '$k$-$\epsilon$' closure scheme using the generic length scale turbulence model (Umlauf and Burchard, 2003). Further details of the model have been documented in Chen et al. (2021).

### 2.2 Observational data

In-situ data were acquired by two different methods. From 23 July to 1 August 2018, a field measurement was conducted at the Dogger Bank (Figure 1) during the Poseidon cruise POS526. This was part of a multidisciplinary research initiative of GEOMAR Helmholtz Center for Ocean Research (GEMOMAR, 2019). During the cruise, high-resolution temperature and salinity data were sampled by CTD (Sea-Bird SBE 49 FastCAT) measurements. Long-term (interannual) data were further obtained from two fixed automatic oceanographic platforms in the North Sea: Nordseeboje III (NSB III, $54°41'$N and $6°47'$E) and FINO-1 ($54°00.892'$N and $6°35.258'$E)(see Figure 1 for the locations). They are part of the Marine Environmental Monitoring Network system (MARNET), operated by the German Federal Maritime and Hydrographic Agency (Bundesamt für





Seeschifffahrt und Hydrographie, BSH). On these MARNET platforms, water temperature and salinity at different depths are
collected continuously and processed to hourly intervals.

A 38-year time series SST over the 1982-2019 period was obtained by the European Space Agency Sea Surface Temperature Climate Change Initiative (ESA SST CCI) Level 3 products (1982-2016) (Merchant et al., 2019) and the Copernicus Climate Change Service (C3S) Level 3 product (2016-2019). These daily-mean datasets cover the European Northwest Shelf Ocean with a spatial resolution of 0.05 degrees by 0.05 degrees. The data were retrieved from the Copernicus Marine Service
(https://resources.marine.copernicus.eu/).

## 2.3   Temperature analysis

Each calendar year was divided into two separate seasons, i.e., the cooling season and the warming season, according to the trend of air temperatures in the annual variation. Normally, the cooling season lasts from January to mid April and then from September to December, while the warming season is from mid April to August. In cooling seasons, cold-spells may be present
when there are periods of at least 5 consecutive days in which the temperature is lower than the threshold of the 10th percentile (Wakelin et al., 2021). In the warming season, MHW events are identified, following the criteria introduced by Hobday et al. (2016), i.e., when the water temperature exceeds the threshold of the 90th percentile within in a period of at least 5 consecutive days.

## 2.4   Quantification of stratification

As introduced by Simpson (1981), the potential energy anomaly $\phi$, is frequently used as a suitable measure of the degree of stratification. This variable indicates the amount of mechanical energy (per $m^3$) required to instantaneously homogenise the water column with given density stratification. The parameter $\phi$ is defined as follows:

$$\phi = \frac{1}{D} \int_{-H}^{\eta} gz(\overline{\rho} - \rho)\mathrm{d}z, \tag{1}$$

in which

$$\overline{\rho} = \frac{1}{D} \int_{-H}^{\eta} \rho\mathrm{d}z \tag{2}$$

is the vertical mean water density and $g = 9.8 \text{ m s}^{-2}$ is the gravitational acceleration. The instantaneous total water depth is given by $D = \eta + H$, with $\eta$ and $H$ being the sea surface elevation and the time mean water depth, respectively. Note that the water density $\rho$, which is not part of the standard NEMO model output, was calculated (at 1 atm) following Millero and Poisso (1981) (details are provided in Appendix A).





To quantify the vertical structure of stratification and evaluate its suitability, the gradient Richardson number, $Ri$, is computed:

$$Ri = \frac{N^2}{S^2},$$ (3)

with the definition of the buoyancy frequency

$$N = \left(-\frac{g}{\rho}\frac{\partial \rho}{\partial z}\right)^{1/2},$$ (4)

and the vertical shear

$$S = \left[\left(\frac{\partial u}{\partial z}\right)^2 + \left(\frac{\partial v}{\partial z}\right)^2\right]^{1/2}.$$ (5)

Here, $u$ and $v$ are the horizontal velocity components in m s$^{-1}$ obtained from the model in the same time intervals as those of temperature and salinity.

## 3    Results

The model results were compared with the in-situ measurements obtained by multiple operating systems (Figure 2). The model was capable of capturing the main features of the observation. At MARNET station NSB III, which is located in the centre of the eastern part of the southern North Sea (see Figure 1), the model data followed the seasonal cycles of the temporal variations in the observed air temperature from 2011 to 2018. In the water, the difference between the observations and the model usually occurred during the warming period of each year (May to September). The interannual cycle of the water temperature followed

the interannual variation in air temperature above the sea surface. However, the intensity of the water stratification showed no obvious seasonal pattern. Its appearance was more related to air temperature changes. For example, large stratification occurred during the summers of 2014 and 2018, when the surface-to-bottom temperature difference was much larger than that during the periods before and after. In 2015 and 2017, the surface-to-bottom temperature difference was unapparent, and fluctuations in air temperatures were small.

Figure 3 further compares the model data and observations at different locations of the southern North Sea in July 2018. At the Dogger Bank and NSB III sites, the model reproduced high temperatures within a thin layer (approximately $5 \sim 10$ m) near the surface. The temporal change in water temperatures mainly occurred in the upper water layers (above 20 m depth). Below 20 meters, the water temperature was relatively stable, with values approximately 10°C. The differences between the surface and the bottom reached more than 15°C in both the model and the observations. At FINO-1, the difference between

the modelled water temperature and the in-situ measurements was nearly non-existent. Inconsistencies between the model and

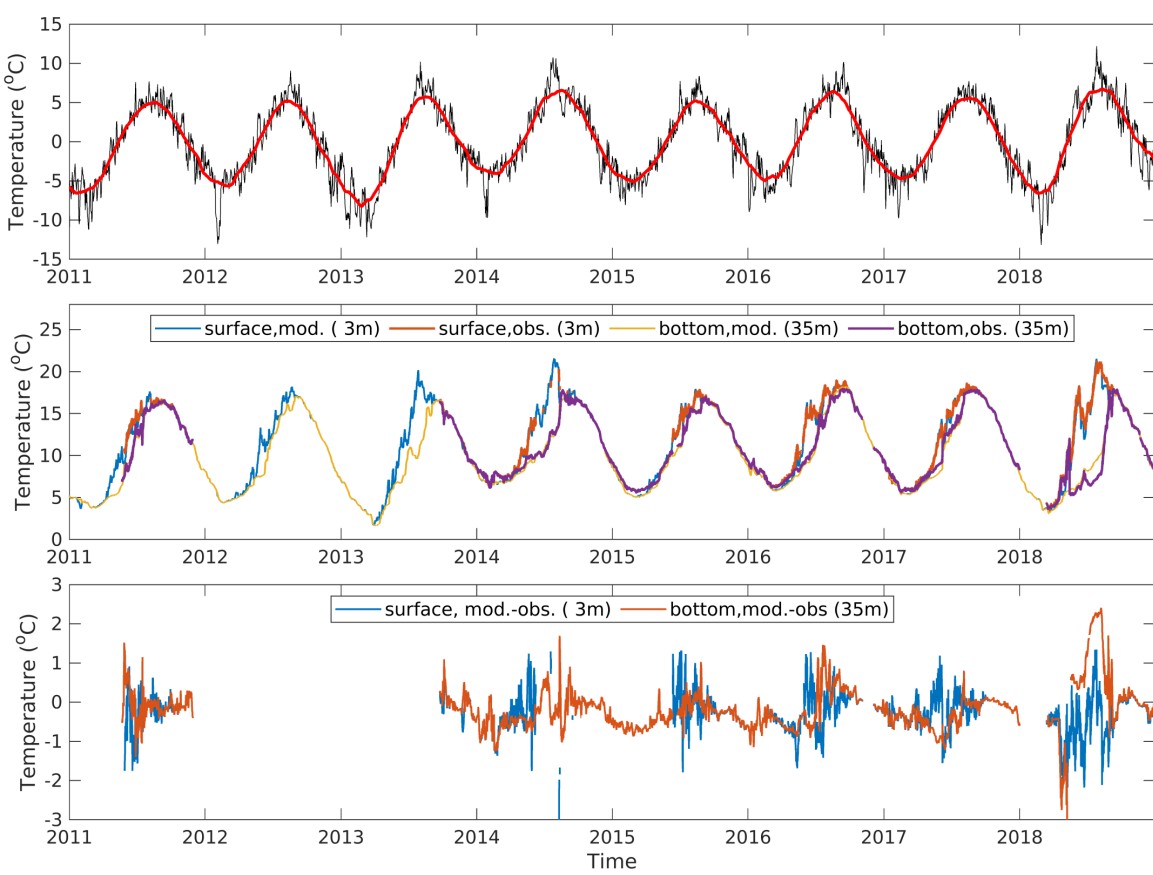

**Figure 2.** Upper panel: Interannual variation in the air temperature (multiyear mean removed) at NSB III. The black line is the daily resolution, and the thick red line is the 3-month moving averaged data. Middle panel: At the same location, the interannual variation in the seawater temperature at the surface and the bottom from the observations (obs.) and the simulations (mod.). Bottom panel: The differences between the modelled seawater temperature and the observed temperature at the surface and the bottom.


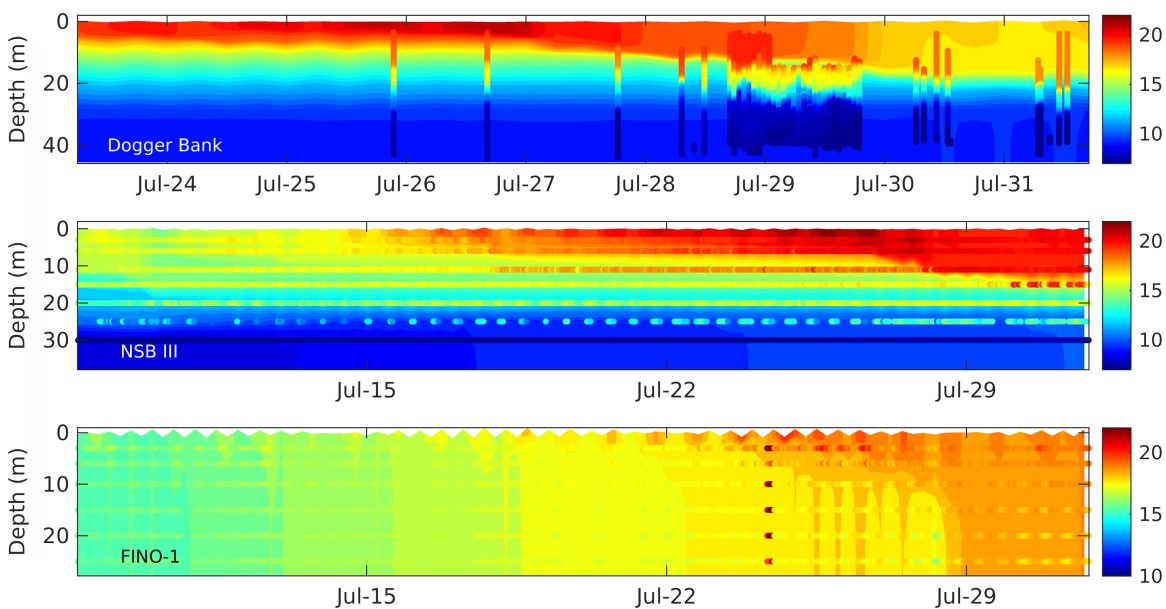

**Figure 3.** Water temperatures (in °C) of the in-situ observations and the model simulations in July. In the upper panel, observations are taken by the CTD profiler during the Poseidon campaign and in the middle and lower panels, the dotted lines are MARNET data measured at fixed water layers.

observations were also observed. The model overestimated the observations at deeper layers. Below the 20 m, e.g., at Dogger Bank and NSB III, the model presented temperatures of approximately $1 \sim 3$°C higher than the in-situ data.

Figure 4 shows an annual variation in SST at the NSB III platform in 2018. The modelled SST was compared with the satellite reprocessed data. The features of the SST annual variation at other stations were similar to those of NSB III and are not shown. The model represented the satellite data from January to May and mid-August to December, with an error less than 1°C. Differences between the two dataset were mainly found from June to August, where fluctuations in the satellite reanalysed data were much smaller than those in the model. One reason could be the smoothed measurements due to gridding and gap-filling of level 3 data. Marine heatwave events in the North Sea region were detected with the 90th-percentile threshold (Hobday et al., 2016), obtained from 38-year time series SST statistics. In 2018, two MHWs were detected at NSB III: May 24 to June 28 and July 8 to August 5. Note that May $4 \sim 15$ was another period of intensive SST incline. However, it was not defined as an MHW event because of the relatively low temperatures.

Regarding the relationship between air temperature variations and the occurrence of seasonal stratification, further comparisons were made between a typical normal year, i.e., 2015, the extreme heat wave year of 2018 and multiyear means (Figure 5). In the warming period of 2018, there were three periods in which air temperature was higher than the multiyear mean. The maximum air temperature anomaly reached 6°C. Correspondingly, three 'heat spikes' were present on the SST curve. During each period, the SST sharply inclined, whereas the water temperature in the deep layers hardly changed. In 2015, 'heat spikes'

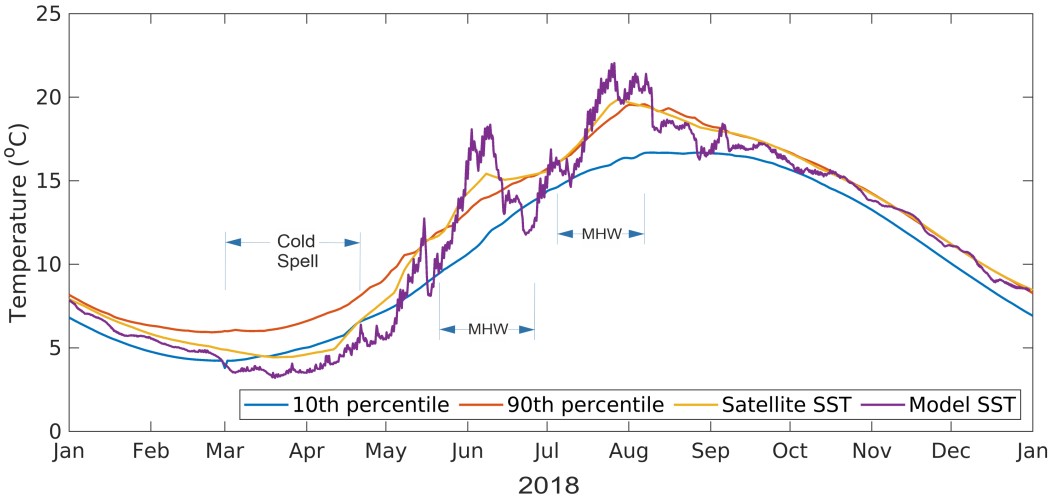

**Figure 4.** SST (in °C) at the NSB III platform of the satellite data, the model simulations in 2018 and the 10th/90th-percentile SST of the multiyear. One cold-spell and two MHWs are detected.

were present in the SST associated with a rapid air temperature increase from June to July. However, contrary to the summer of 2018, the summer of 2015 was colder than the average during the warming season, with the maximum air temperature anomaly being 3°C lower than the multiyear mean.

In the absence of turbulent mixing, high air temperatures (warmer summer) lead to a high SST and intensify the temperature differences between the local sea surface and the bottom. This can be seen in the three 'heat spike' periods in 2018. Note that in 2018, the first 'heat spike' was not identified in an MHW event due to the relatively low temperature. In 2015, no atmospheric heatwave or MHW events were detected. Correspondingly, the SST was below the average, and the surface-to-bottom temperature difference was smaller than the multiyear mean.

The longer memory of seawater than that of air to low temperatures leads to larger and more stable temperature stratification. As shown in Figure 5, at the end of February to middle March 2018, there were two periods of 'cold waves' with air temperatures 7°C lower than the multiple-year mean. As a response, the seawater was colder than usual, causing the bottom temperature to be 2°C lower than the average. This low-temperature signal remained much longer in the water than in the atmosphere and yielded colder bottom water over the entire warming season.

The monthly mean spatial distribution of the potential energy anomaly ($\phi$) in the southern North Sea in 2018 (Figure 6) provides an overview of the seasonal cycle of water stratification development during the extreme temperature conditions. The water become stratified by April in the northern part of the domain, i.e., the north side of the 50 m isobath. From May to August, the stratification developed in the area north to 54°N latitude. South of the 50 m isobath, the potential energy anomaly reached $400\,\mathrm{J\,m^{-3}}$ on average from July to early August, with a maximum value of approximately $800\,\mathrm{J\,m^{-3}}$ in the area

between Dogger Bank and MARNET station NSB III. In September, the $50\,\mathrm{J\,m^{-3}}$ isoline retreated to the north of the 50 m



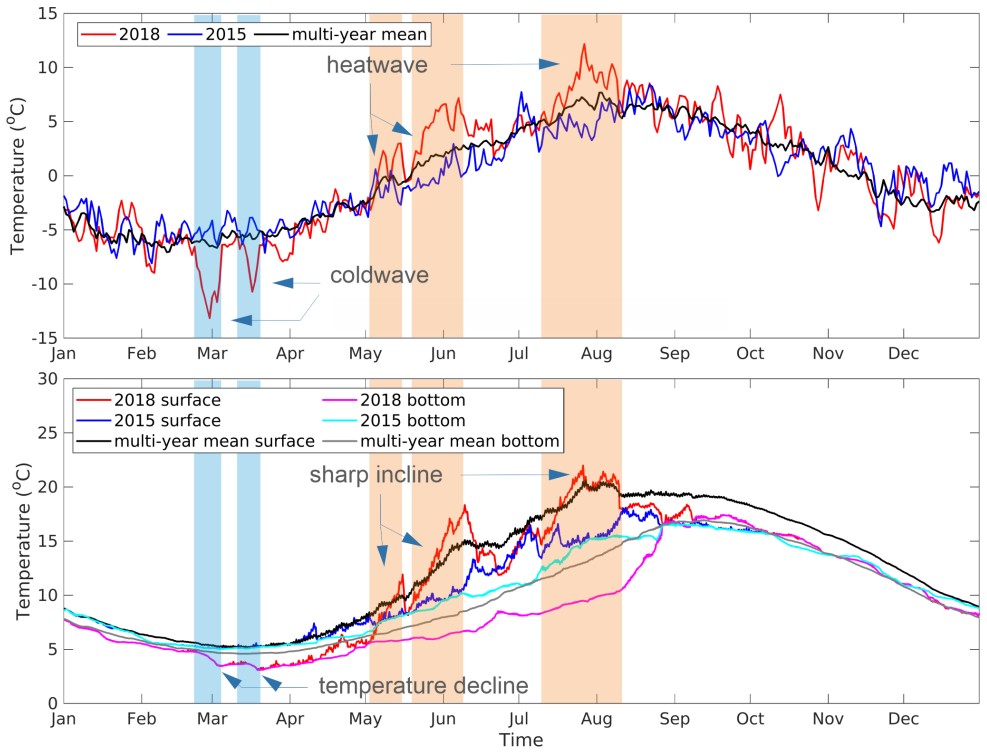

**Figure 5.** Upper panel: Annual variation in the simulated air temperature (multiyear mean removed) at NSB III for 2018 (red line), 2015 (blue line) and multiyear mean (black line). The periods in which air temperature increased rapidly during the warming season of 2018 are demonstrated with orange frames. Likewise, the periods in which air temperature decreases rapidly in early spring are demonstrated with blue frames. Bottom panel: Similar to the upper panel but for the simulated seawater temperature at the surface and the bottom.

depth line. South of 54°N latitude, the water column was mostly well mixed. The stratification near the Dutch coast and the German Bight was due to river runoff.

To illustrate inter-annual variability of the water stratification in the southern North Sea, the potential energy anomaly of each year to the multi-year meant (2011-2018) is computed by averaging for three months from June to August (the main stratified period, see Figure 6). The results are shown in Figure 7. In the years 2013, 2014 and 2018, the stratification is stronger than the average, whereas, in the other years it is weaker. In the area the water depth is less than 50 meters, the largest inter-annual variation occurs between 3-8°E longitude and north to 54°N latitude. The mean potential energy anomaly is $300 \sim 400$ J m$^{-3}$ higher than the multiyear mean in 2018, while it is $200 \sim 300$ J m$^{-3}$ lower than average in 2015. Note that the largest inter-annual variations of the stratification occur in the east part of the southern North Sea.


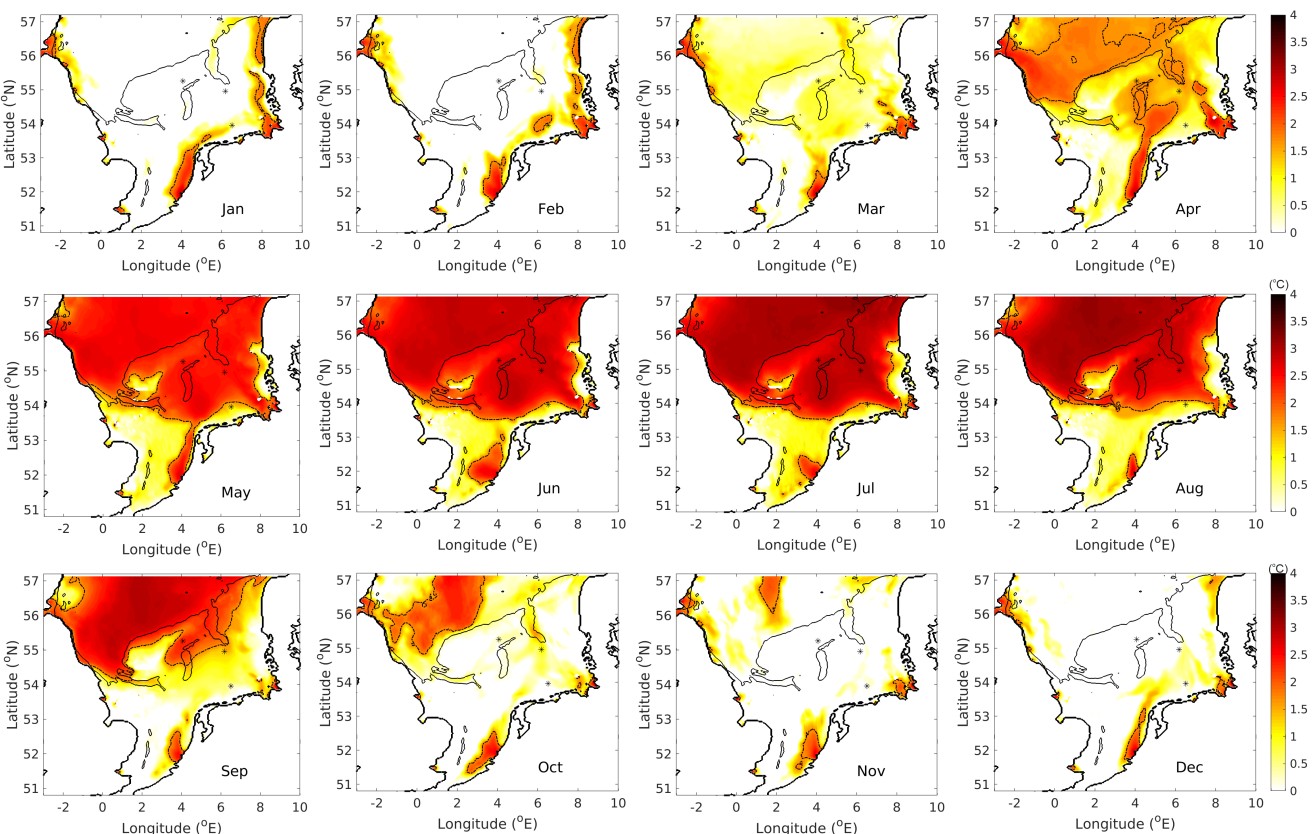

**Figure 6.** Evolution of the potential energy anomaly $\phi$ (see equation 1, unit: J m$^{-3}$ in log10 scale) in 2018. Dashed line indicates $\phi = 50$ J m$^{-3}$. The water column is considered to be stratified when $\phi$ is above this value. Thin black lines indicate the location of 50 m depth.

## 4 Discussion

As illustrated in Figures 6 and 7, the stratification/destratification process in the North Sea was temporally and spatially dependent. The data analysed at a single point was incapable of revealing a complete image of the relationship between the seasonal thermal stratification and the varying air temperatures in the North Sea. van Leeuwen et al. (2015) investigated the physical conditions in the North Sea, and attributed regimes of different types of stratification to thermal-induced, salt-induced, river-induced and turbulent tidal mixing. They found that in the southern North Sea, a large area could not be characterised by a dominant physical mechanism. With the focus on thermal-induced stratification, we analysed the relation between the variation in air temperature and the seasonal thermal stratification in the North Sea. A coefficient $R$ that indicates a linear correlation between the air temperature and the potential energy anomaly for the summer of 2018 was computed and mapped, as shown in Figure 8. The coefficient $R > 0.8$ was obtained mainly in the middle and northern parts of the North Sea (Figure 8a). This area was consistent with the regime of seasonal stratification identified in van Leeuwen et al. (2015). Closer to the coast, a lower $R$ was obtained. Five locations in the southern North Sea were selected regarding the different values of $R$ (see Figure 8a) and

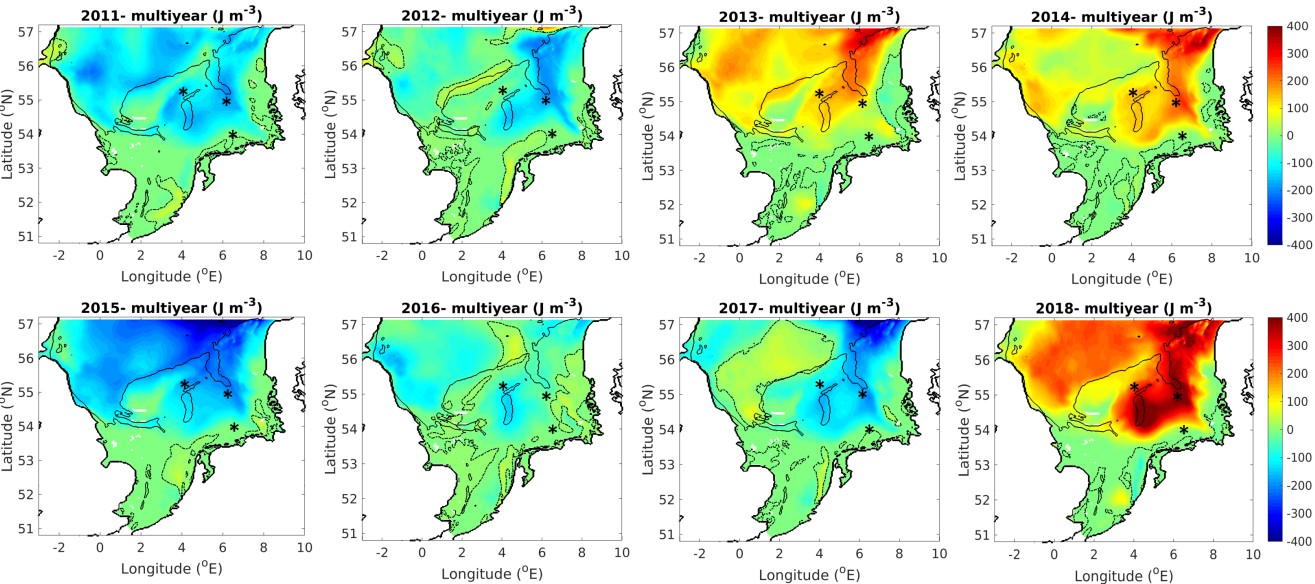

**Figure 7.** The difference in the three-monthly mean (June, July, August) potential energy anomaly (J m$^{-3}$) between a specific year and the multiyear mean. Dashed line indicates $\phi = 50$ J m$^{-3}$. Thin black lines indicate the location of 50 m depth.

scattered based on the air temperature versus potential energy anomaly (Figure 8b). Note that both the Dogger Bank and the NSB III are stations located in the regime for which the stratification type was unclassified in van Leeuwen et al. (2015). A linear correlation existed between the increase in air temperature and the increasing stratification, with $R = 0.8$ at Northsea

Mid and $R = 0.7$ at Dogger Bank. The larger potential energy anomaly in the middle and northern North Sea (see 'Northsea Mid' in Figure 8a) was due to the deeper water depth. At NSB III, $R = 0.5$ also illustrates a linear increase in stratification with air temperature. At the Dutch Coast and the Norwegian Trench, $R$ was negative. For the former location, the water column was stratified due to river runoff, and for the latter, the water was permanently stratified. The potential energy anomaly was uncorrelated with the air temperatures at the two locations.

Mapping the correlation between the air temperature and the potential energy anomaly for different years yielded similar spatial patterns and values of $R$ in the North Sea except for the middle part of the southern North Sea, where lower $R$ values were found for the years with relatively colder summers. For example, in 2015, $R$ fell by $0.1 \sim 0.3$ in the area between Dogger Bank and NSB III (not shown), indicating a much lower correlation between summer stratification and air temperature.

Similar to that of Figure 4, MHW events for each warming season (May 1 to August 31) of the simulation period 2011$\sim$2018

were detected, and the number of MHW days ($\mathcal{M}$) was counted. Then, the changes in $\mathcal{M}$ with respect to its multiyear mean $\sum_{i=1}^{n} \left| \mathcal{M}_i - \overline{\mathcal{M}_n} \right|$ was computed for the modelling period. Here, "n = 1, 2, 3, ..." counts the number of years. The overline ($\overline{\cdot}$) denotes the multiyear mean. Similarly, the number of days ($\mathcal{N}$) that the water column was stratified ($\phi > 50$ J m$^{-3}$) and the

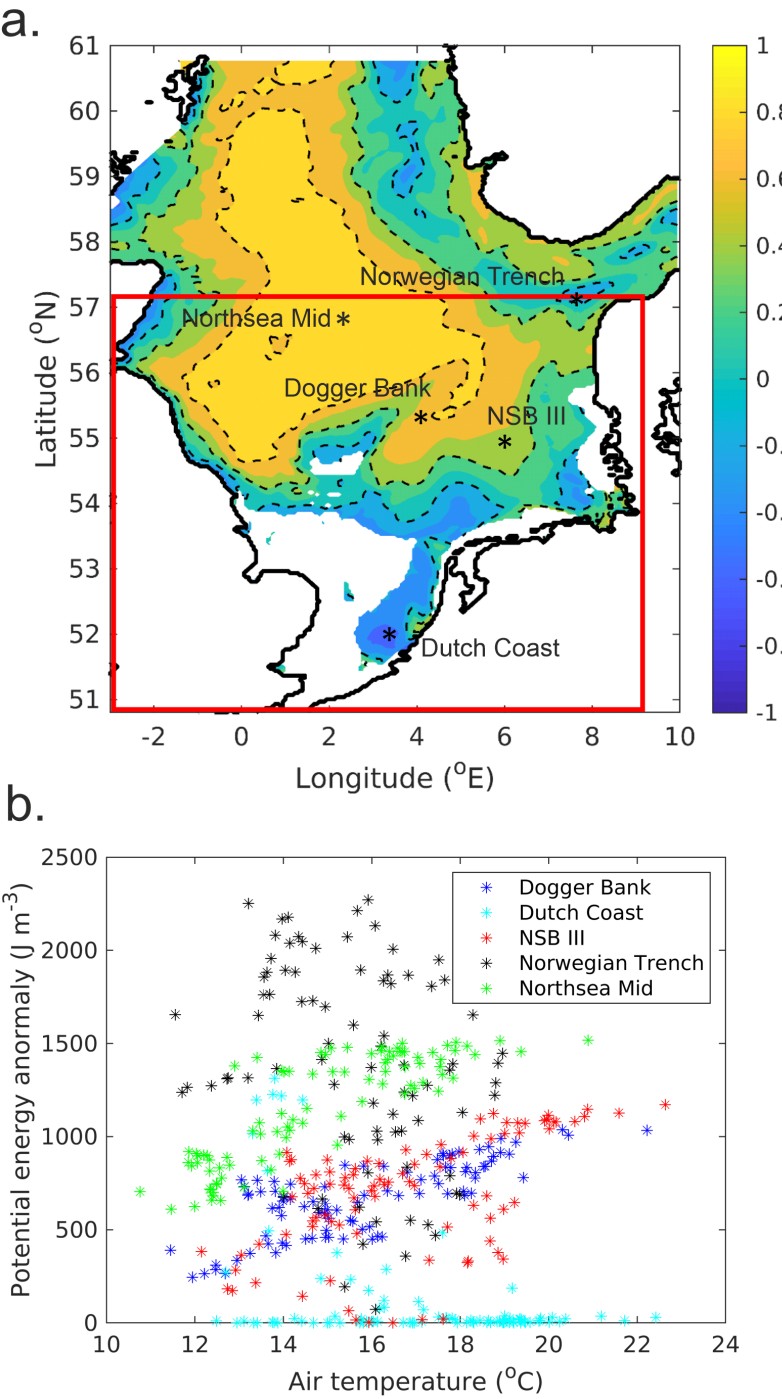

**Figure 8.** (a): Computed correlation coefficient 'R' between the air temperature and the potential energy anomaly for summer 2018 (June-August). Note that the area with no stratification ($\phi < 50\,\mathrm{J\,m^{-3}}$) is cut off. Dashed contours indicate $R = 0, 0.4$ and $0.8$. (b): The relationship between the air temperature and the potential energy anomaly in the different regimes of the North Sea. The locations are illustrated in (a). The red frame indicates the region of the southern North Sea.




changes in stratified days $\sum_{i=1}^{n} \left| \mathcal{N}_i - \overline{\mathcal{N}_n} \right|$ were computed. The sensitivity of stratification to the occurrence of heatwaves was quantified with the ratio between the varying heatwave days and varying water-stratified days:

$$r = \frac{\sum_{i=1}^{n} \left| \mathcal{N}_i - \overline{\mathcal{N}_n} \right|}{\sum_{i=1}^{n} \left| \mathcal{M}_i - \overline{\mathcal{M}_n} \right|}. \tag{6}$$

Note that the region where no MHW events were detected ($\mathcal{M} = 0$) was not considered. Figure 9 shows the spatial distribution of $r$ in the North Sea. The value of $r$ varies between $0 \sim 1$. When $r = 0$, the number of days that the water column is stratified does not change interannually. This pattern occurs for both permanently stratified regions and permanently mixed regions. The former was found in the Norwegian Trench and the latter was mainly found in the west part of the southern North

Sea, as well as in the shallow shoal of the Dogger Bank, along the Danish coast and the southern part of the German Bight. If the ratio $r = 1$, i.e., the changes in the number of water stratification days equal the number of MHW days, it illustrates the dependence of the thermal stratification on the occurrence of MHWs. In the region the water depth was greater than 50 m $r < 0.2$. This region was consistent with the region where $R > 0.6$, where stratification seasonally occurs with annual varying air temperature. South to the 50 m depth, $r > 0.3$ and increased towards the coasts, indicating an enhanced influence of MHWs

on water stratification. The region where $r$ reached the maximum was between $6.5°E$ and the Danish coast.

Large values for $r$ were also observed between the MHW-induced stratification region and the permanently mixed region and near the UK coast in the northern North Sea. These regions are shallow and mostly mixed by tides. The water stratification is due to the short and intensive increase in air temperature. As shown in Figure 5, three periods of stratification ($\phi > 50$ J m$^{-3}$) were detected. Similar to the counting of MHW days, we counted the number of days from May to August of 2011-2018

in which the air temperature sharply increased. In the present study, the criterion was set to $0.2°C/day$, i.e., the air temperature increase rate was at least two times larger than the warming period averaged increase rate of the multiyear mean. After applying Eq. 6, a ratio between the changes in intensive air temperature increases and the changes of water stratification was obtained (see Figure 10). Overall, the spatial distribution of the ratio between the changes in intensive air temperature increases and the changes in water stratification was similar to that of the ratio between the changes in MHW days and the stratification days.

This was particularly found in the area adjacent to the tidal mixing zone and the MHW-induced stratification zone. The region was consistent with the intermittently stratified domain characterised by van Leeuwen et al. (2015).

The sensitivity of thermal stratification to summer heatwaves was related to changes in water depth. Figure 11 presents the annual variation in the gradient Richardson number $R_i$ for the selected stations with different depths in the regime where $R \geq 0$ (see Figure 8). The water column was considered stably stratified when $\log_{10}(R_i) \geq 1$, i.e., the buoyancy was an order of

magnitude larger than the vertical shear. For both 2015 and 2018, the water column became stratified from May to September in the middle of the North Sea, with a comparable value of the gradient $R_i$. While approaching the coast, the gradient $R_i$ decreases with the reduced water depth. At Fino-1, the water column was mostly well-mixed, despite some intermittent stratifications on the time scale of $1 \sim 2$ days. Differences between 2015 and 2018 were observed at Dogger Bank and NSB III. At Dogger

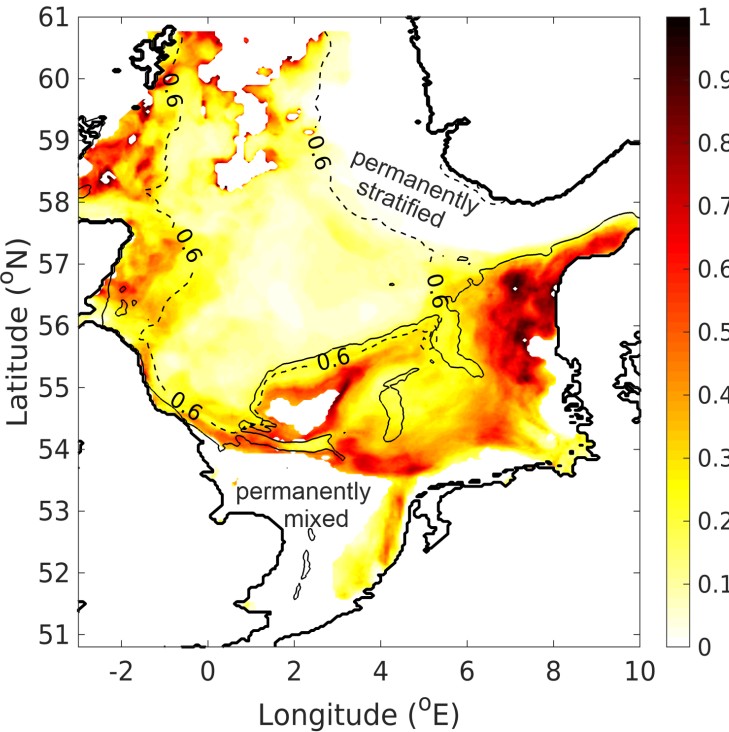

**Figure 9.** The ratio between the number of days when the water was stratified and the number of days with MHW events. The dashed contour line corresponds to the correlation coefficient '$R = 0.6$' between the air temperature and the potential energy anomaly for the summer time of the multiyear mean.

Bank, stratification occurred from June to the end of August in 2015, which was both shorter and less intensive than the period in 2018. At NSB III, stable stratification existed only in 2018.

Further analysis of the vertical distribution of the gradient $R_i$ revealed that the maximum value of $R_i$ was obtained in the middle of the water column. At Northsea Mid, it was located approximately 35 m in depth. Below 45 m, $\log_{10}(R_i) \sim 0$ and the water column was homogeneous throughout the entire year. At NSB III, the period in which the large $R_i$ values occurred was consistent with the occurrence of MHW events. In 2018, $\log_{10}(R_i) \geq 1$ during June and from early July to August. Moreover,

the Dogger Bank and NSB III that the maximum $R_i$ shifted from the upper water layers down to the bottom with the value increased during the early stage of the warming season. However, in the late warming season (July and August), the maximum $R_i$ remained in the middle water layers. At both Dogger Bank and NSB III, the water column below the 25 m depth was homogeneous throughout the entire year. At Fino-1, the water depth (25 m) was too shallow to maintain long-term stable stratification.

Note that the present study takes into account the impact of ocean waves on circulation. Such an impact usually results from different processes, e.g., turbulence due to breaking/nonbreaking waves and the transfer of momentum from breaking waves to

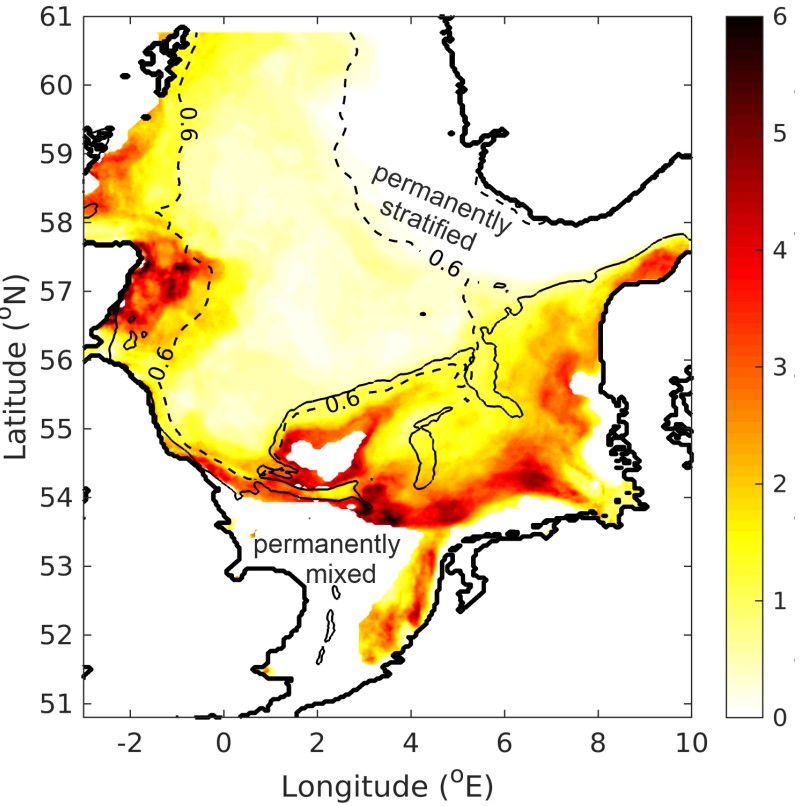

**Figure 10.** The ratio between the number of days when the water is stratified and the number of days in intensive air temperature incline. The contour line with a value of 0.6 indicates the area with a correlation coefficient '$R = 0.6$' between the air temperature and the potential energy anomaly for the summer time of the multiyear mean.

currents (Davies et al., 2000; Babanin, 2006; Breivik et al., 2015). The latter is important in shallow waters such as the Baltic (Alari et al., 2016) and the North Sea (Staneva et al., 2017; Wu et al., 2019). By including wave-induced turbulent mixing, the simulation was capable of resolving the turbulent mixing length and modelling the vertical stratification and circulation in the
North Sea (Staneva et al., 2017, 2021), especially near the coastal zones. For further details of the wave impact, refer to the Supplemental materials for results of the model that was uncoupled with WAM.

Apart from circulation-wave coupling, stratification itself also influenced circulation. On a time mean momentum budget, where baroclinic forcing balances the internal friction, turbulent eddy viscosity decreases while turbulence is eliminated by the increasing degree of stratification. Baroclinic circulation is more pronounced with stronger current velocities (larger vertical
shear) in warmer climate conditions. Huang et al. (1999) modelled the seasonal thermal stratification and baroclinic circulation in the Bohai Sea, where the water depth is shallow (with a maximum of approximately 70 m). They showed an enhancement of water transport by baroclinic circulation from a weakly stratified situation in spring to a strongly stratified situation in summer. In the North Sea, Lwiza et al. (1991) conducted a study of thermal-induced circulation along tidal mixing fronts in
**Figure 11.** Gradient Ri for 2015 (left column) and 2018 (right column) at Northsea Mid, Dogger Bank, NSB III and Fino-1 on a log10 scale.

summer. Luyten et al. (1999) further analysed the cycle of thermal fronts and baroclinic circulation in the North Sea at different
ranges of timescales. They showed the increase/decrease in vertical mixing with the enhancement/reduction of the depth mean
temperature in summer through the surface heat flux. Moreover, decreased vertical turbulent mixing reduced the magnitude
and horizontal shear of the baroclinic currents. This was found to be important feedback from turbulent mixing to the frontal
temperature gradients and baroclinic circulation (Luyten et al., 2003). These mechanisms would be pronounced in the case of
the extreme events. Schrum et al. (2003) identified unusual stratification and volume transport in the 1990s by analysing 40-year
data. However, the study was still in a preliminary stage and with a focus on decadal timescale wind impacts on the exchange
flow at the open boundaries. Recently, Chen et al. (2021) investigated the heat budget of the North Sea. They demonstrated that
modifying water temperatures by assimilating satellite SST had an impact on the water circulation on an annual timescale and
further affected advective heat transport in the North Sea. Comparatively, extreme events, i.e., summer heatwaves, had more
intensive influences on SST but within a relatively shorter timescale. Exploring the reaction of the regional water circulation to
extreme events in the southern North Sea is an interesting topic and deserves further study.





## 5    Conclusions

This study investigated the influence of extreme climate events, i.e., European heatwaves, on the occurrence of summer stratification in the North Sea. With a numerical model that simulated the water temperatures in multiple years, the work addressed the question remaining after the study by van Leeuwen et al. (2015): What kind of 'interannual variability' results in the failure
of classifying one-third of the North Sea as one of the defined stratification regimes?

Based on the model simulations, a potential energy anomaly was calculated for multiple years in the North Sea domain. In the southern North Sea, stratification developed from May to August. The intensity of this stratification presented an obvious interannual variation that was related to the occurrence of summer heatwave events. The data from 2018 were analysed as a case and compared with those of the multiple-year mean and the normal year 2015. Regarding the main factors that affect the
intensity and duration of the thermal stratification in the southern North Sea, the comparison of temperature data near the water surface and the bottom revealed three aspects. The first was the intense increase in SST in a short time and the second was a relatively higher water temperature during summer compared to the multiple-year mean. The third was the memory of the near-bottom water layer to the low air temperatures during early spring.

By computing the correlation coefficient ($R$) between the air temperatures and the potential energy anomaly, we identified
the region of thermal stratification in the North Sea ($R > 0$). The region covered north of 54°N of the North Sea, excluding the permanently stratified Norwegian Trench and the permanently well-mixed coastal seas near the Danish Wadden Sea. Features in stratification were different between the north and south sides at a 50-m depth. Contrary to the northern part, where the water column was seasonally stratified every year, the southern part was stratified in the years when heatwaves occurred. The dependence of thermal stratification on heatwave events was quantified by the ratio $r$ between the change in water stratification
days in 2011-2018 and the change in MHW days during the same period. The ratio $r = 1$ indicates that the change in thermal stratification days is the same as the change in MHW days. Large ratios ($r > 0.8$) were found on the southern side of the 50 m isobath between 6.5°E and the Danish coast, which covered most of the unclassified stratification regime of van Leeuwen et al. (2015).

Water depth appeared to be the factor that controlled the sensitivity of the stratification to the summer heatwave. Apart from
the coast, the stratified period was more dependent on the number of days a heatwave event lasted. After analysing the structure of the gradient Richardson number $R_i$, it was found that heatwave-induced seasonal stratification mainly occurred in the region where the water depth was between 35 m and 50 m.

This research is the first to link heat wave events with the occurrence and persistence of density stratification in the southern North Sea. In a broader context, this research will have fundamental significance for further discussion of the secondary effects
of heat wave events, such as in ecosystems, fisheries, and sediment dynamics. With the growing debate regarding the impacts of increasing extreme climate events, assessments of these factors will be of more importance in the North Sea in the future.

*Code availability.*   The model code is available at https://www.nemo-ocean.eu/for NEMO and https://github.com/mywave/WA M for WAM.


*Data availability.* The outputs of GCOAST simulations (in NetCDF format) are available upon request to the corresponding author.

## Appendix A: Estimates of the oceanic water density

The water density $\rho$ in Eq. 2 is given by

$$\rho(S,T) = \rho_r + \mathrm{A}S + \mathrm{B}S^{\frac{3}{2}} + \mathrm{C}S^2. \tag{A1}$$

In this equation, $S$ is the salinity of sea water in ppt (parts per thousand by volume). The reference density $\rho_r$, the coefficients A, B and C are also functions of temperature $T$ in $^oC$ with expressions given by Millero and Poisso (1981):

$$\rho_r = 999.842594 + 6.793952 \times 10^{-2}T - 9.095290 \times 10^{-3}T^2$$
$$+ 1.001685 \times 10^{-4}T^3 - 1.120083 \times 10^{-6}T^4$$
$$+ 6.536332 \times 10^{-9}T^5;$$
$$\mathrm{A} = 8.24493 \times 10^{-1} - 4.0899 \times 10^{-3}T + 7.6438 \times 10^{-5}T^2$$
$$- 8.2467 \times 10^{-7}T^3 + 5.3875 \times 10^{-9}T^4;$$
$$\mathrm{B} = -5.72466 \times 10^{-3} + 1.0227 \times 10^{-4}T$$
$$- 1.6546 \times 10^{-6}T^2;$$
$$\mathrm{C} = 4.8314 \times 10^{-4}.$$

Here $S$ and $T$ are obtained from the model with a temporal resolution of 6-hours. Moreover, a 6-hour resolution model output of the sea surface elevation $\eta$ is taken for computation (see the main text eq. 1 and 2).

*Author contributions.* Wei Chen wrote the article, analysed and interpreted the results. Wei Chen and Joanna Staneva conceived the study. Se-
bastian Grayek ran the simulations and responsible for all technical issues of the modelling. Joanna Staneva and Johannes Schulz-Stellenfleth contributed to the writing. Jens Greinert performed the field works and collected observational data.

*Competing interests.* The authors declare that they have no conflict of interest.

*Acknowledgements.* This study is supported by the Digital Earth project (funded by the Helmholtz Association). The authors gratefully acknowledge the German Climate Computing Centre (DKRZ) for providing computing time on the Supercomputer MISTRAL.





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
