# Peer review of "The role of heat wave events on the occurrence and persistence of thermal stratification in the southern North Sea"

_Natural Hazards and Earth System Sciences, 2021_

## Referee Comment (RC2)

**The role of heat wave events on the occurrence and persistence of thermal stratification in the southern North Sea**

**Review**

The manuscript aims to investigate the contribution of marine heatwaves (MHW) to thermal stratification in the North Sea by using ocean-wave coupled numerical experiments performed during a 8-year period.

Comparing with observations (in-situ and remote-sensing), they observed large stratification events in the summers of 2014 and 2018, when large surface-to-bottom differences in water temperature co-occurred with high air temperatures.

They argue that in absence of turbulent mixing high air temperatures lead to high SST and intensified stratification, triggered also by the long memory of seawater compared to air to low temperatures in the cooling season (e.g. January-April).

Thermal stratification and MHWs show a well-defined dependency in areas, along the Danish coast, which were not well characterized in the literature as one of the density stratification regimes in the North Sea.

The vertical structure of stratification (Ri) shows the water depth as the factor modulating the sensitivity of the stratification to the summer heatwave (max values observed in the middle of the water column).

The work is well developed and well placed within the framework of the scientific open issues about the effect of extreme climate events. The authors use robust analysis and provide an exhaustive physical interpretation of all the results obtained, which address the goals of the study. The publication is recommended.

Here follow some suggestions, which could help improve certain parts of the manuscript.

**General Comments**

1. The sensitivity of ocean-wave coupling is addressed in supplementary materials (SM), but either a dedicated Appendix on the wave-induced processes included in the coupling, or an additional Figure in SM on the sensitivity to the wave-induced turbulent mixing, could help the reader to fully understand the results (e.g. Lines 270-275).

2. The quality of the graphics and Figure captions could be improved (look at specific comments).

**Specific Comments**

**Introduction**

**Line 17:** the reference IPCC, 2012 should be updated to AR6

**Line 26**: "In the North Sea …" - rephrase.

**Line 55:** "who showed…" – rephrase.

**Line 56** "Considering the interface …" – rephrase.

**Section 2**

**Line 99:** Check the reference GEMORAR, 2019

Line 102: omit "for the location".

**Section 3**

**Line 158-159:** rephrase. Something like: "The results were further compared with satellite data. Figure 4 shows an annual variation in the modelled and remote sensed SST at the NSB III platform location in 2018.

**Line 194:** multi-year mean.

**Section 4**

**Line 232:** "….the number of days when the water …"

**Line 265:** "…down to the bottom": it is hard to distinguish max Ri values at the bottom, and the statement is in contrast with the sentence at Lines 267-268. Consider rephrasing.

**Line 277-295:** It could be shorter and move to the conclusions.

**Comments about Figures**

**Figure 2:**
The bottom panel should be either described in the manuscript or removed.

Caption
**"**Bottom panel: the differences …".

**Figure 3**
The text in the caption should be improved.

**Figure 4**
The text in the caption should be improved.

**Figure 6**
Check the unit (see palette).  The text in the caption should be improved describing clearly what is displayed on the background. It is hard to distinguish between dashed and solid black lines.

**Figure 9**
What about the solid line? See relevant comment for Figure 6.

**Figure 10**
See previous comments.

---

## Author Comment (AC1)

**Reply to the comments of reviewer 1 on the NHESS submission:**

**'The role of heat wave events on the occurrence and persistence of thermal stratification in the southern North Sea'**

*We would like to thank the Editor and the reviewer for their careful reading of the manuscript and for many constructive comments and suggestions, which were useful to improve the manuscript. Below, we present a point-to-point reply to all comments. The original text of the reviewers is in black and our reply is in italic blue.*

GENERAL COMMENTS

This work provides an interesting insight in the relation between heat waves and thermal stratification in the southern North Sea, by relying on a coupled hydrodynamic-wave model and observational data. The work is flanked by supplementary material containing the results of an uncoupled run. My overall opinion on this manuscript is positive. As general comments, there are two main aspects on which I would like to draw the authors' attention.

1) It is not obvious from the text whether in-situ or potential temperature was used to compute water density for this study. While the latter is the typical choice in pysical oceanography (see for instance Cushman-Roisin and Beckers - Introduction to Geophysical Fluid Dynamics) as potential density contains the actual dynamical information on the water column conditions, the former case would probably need at least some additional discussion. For instance, if in-situ temperature has been used instead of potential temperature (thus providing in-situ density instead of potential density), the results would obviously be affected by water depth with impacts on some of the conclusions (Lines 320-324). Please clarify this point and, if in-situ density has been used instead of potential density, please discuss the choice and its implications.

*We thank the reviewer for pointing out this issue. In this study, the potential temperature is used to compute water. To clarify this point, we revise the sentence (line 120) as 'The reference density $\rho_r$, the coefficients A, B and C are also functions of potential temperature T in $^oC$, …' for the resubmission.*

2) The manuscript and the supplementary material are well written and informative. Nonetheless, I think that including in the manuscript some further comments on the effect of model coupling (besides the mention at L270-276), particularly in terms of the impact of wave description on vertical mixing and heat fluxes at the sea surface and through the water column, would significantly enhance the reach of this work.

[Figure]

Figure 12. Upper panels: The seasonal mean mixed layer depth (MLD) of the coupled NEMO-WAM model run. Lower panels: The relative changes of MLD when comparing the coupled run with the stand-alone NEMO (uncoupled) run. Dotted lines in the lower panels indicate the 0 % contour.

*We appreciate the reviewer's suggestion and added texts in discussion (line 275): "Figure 12 shows the modelled seasonal mean mixed layer depth (MLD) of the coupled NEMO-WAM model run in 2018 and the relative changes when comparing the fully coupled run with the uncoupled run, i.e.,* $(\mathrm{MLD_{coupled}} - \mathrm{MLD_{uncoupled}})/\mathrm{MLD_{coupled}} \times 100\%$*.*

*Following the annual cycle of water temperature, MLD in the southern North Sea decreases from winter to summer and develops again in autumn. The MLD changes quicker in deeper regions than in shallow coastal areas. There is almost no seasonal cycle along with the German Bight coast. The large difference in MLD between the coupled and uncoupled run is found from spring to autumn, when stratification develops with the changing air temperature. In summer, the MLD of the coupled run is approximately 20~40% larger than that of the uncoupled run in the southern North Sea, whereas it is approximately 20% smaller north to the 54ºN. In autumn, the stratification disappears in the southern North Sea, where the MLD differences drop to -10~10%. In the north, where water depth is larger than 50 m, the MLD of the coupled run is approximately 20% larger than the MLD of the uncoupled run. The wave-induced processes change also the heat fluxes at the water surface (Figure 13). The net heat flux is overall positive ( from air to the sea) in spring and summer and from sea to the air in winter and autumn The importance of wave forcing for ocean predictions in the North Sea has been demonstrated by Staneva et al., 2017. The wave-induced processes impact the distribution of the heat*

[Figure]

Figure 13. The same as Figure 12, but for the surface heat fluxes. In the upper panels, positive values denote net fluxes from the air the sea.

*fluxes.   In summer, in the southern North Sea, the fluxes increase by about 20-40% while along the coastal, well-mixed area of the German Bight is the opposite and a reduction of 20 -50% is observed.*

SPECIFIC COMMENTS

L81: Please explain the choice of the study period.

*The study period is 2010-2020.  We have chosen this period because we wanted to analyze,  in addition to the extreme year of 2018, which is a focus of our research,  past several heatwaves and cold spells events.  Besides,  in-situ observations available in this period and used in our study are consistent. Therefore,  we also made model simulations and analyses for both uncoupled and coupled models since 2010.*

L86: Why no other campaigns were considered for model validation?

*Poseidon cruise POS526  data is the only available dataset during the heatwave event within the research area (southern North Sea).   For the whole period, we used also MARNET  in-situ observations and the satellite data.*

L150 and Figure 3: since this is potentially relevant for the conclusion, it could be worthwhile discussing more in detail the mismatch in the profile shape and the modelled anticipation of the cooling at the surface at Dogger Bank (see also General Comment #2).

Based on our investigation, we attribute the premature cooling in the simulation for the Dogger Bank position to a temporal/spatial misplacement due to atmospheric forcing.  Please, note that the atmospheric forcing data used in our simulations have a spatial resolution of 31 km (about half the

distance between the "NSB III" and "Dogger Bank" measurement sites). We postulate that the differences between the observed and simulated profile shapes at the NSBIII and Dogger Bank monitoring sites are also due to the strong lateral mixing. The regional situation, especially in the 2018 heat wave was characterized by strong spatial temperature gradients.

L175-180, Figure 3 and Figure 4: could the range of the modelled heat spikes compared with satellite SST be related with the description of vertical mixing?

The comparisons with in-situ observation of SST demonstrated that simulated temporal variability to be more reliable than the signal in the satellite SST product. As mentioned in the text, this SST product is very smooth. The spikes are due to a combination of a direct response to the changing atmospheric conditions (e.g. abrupt increase or decrease of the air temperature over the shallow area), as well as lateral and vertical mixing in the German Bight area.

L185-190: this result should be discussed in the light of the implications of the mismatch in vertical profile description (Figure 3) for Potential Energy Anomaly

We added in the text a discussion of the Figure: "In general, the model is capable to represent the characteristics of water temperatures in the North Sea. The model errors are small compared to the annual variations in water temperature. One reason for the discrepancies between the model and observation is the spatially and temporally coarse resolution atmospheric forcing data. In July-August 2018, the modelled bottom water temperature is warmer than the observations (Figure 2). During this heatwave period, strong spatial temperature gradients are observed. This leads to additional heat transport from shallower to deeper regions. At the NSB III site, the density stratification in the water column was underestimated by the model, compared to the observations. However, this does not affect the main conclusion of the study, considering that the model error is much smaller than the temperature differences between surface and bottom (10~12 deg.C)."

Figure 5: is there a definition for "rapidly"? If the criterion is the one defined in L245, it could be a good idea to state it also here.

*We appreciate this suggestion. However, in the discussion here, there is no specific definition for 'rapidly'. The orange colour in Figure 5 shows the periods that air temperature is higher than the multi-year mean and SST increases. The blue colour shows the periods that the air temperature is lower than the multi-year mean and SST decreases. With this figure, we qualitatively illustrate the consistency of air temperature changes and SST variation. Definition of 'rapid' air temperature increase is necessary to quantify the relation between air temperature changes and the water density stratification as discussed in detail in section 4. To avoid potential confusion, in the revised manuscript we reformulate the captions in Figure 6 as the following: "In the warming period of 2018, the periods that air temperature is higher than the multi-year mean and SST increases are demonstrated with orange frames. Likewise, in the early spring of 2018, the periods that air temperature is lower than the multi-year mean and SST decreases are demonstrated with blue frames".*

Figure 7: it could be interesting to see also the winter months, in support of the statement at L180.

*Figure 7 shows changes in summer stratification (in terms of potential energy anomaly) of a single year compared with a multi-year mean situation. The results reveal stronger summer stratification in 2014 and 2018 and a weaker summer stratification in 2015.   Similar analyses in winter  (Dec, Jan, Feb) will not illustrate this process,  since the water column is well mixed and density stratification hardly exists in the southern North Sea  (see Figure 6). We find that we did not clearly describe the main message delivered by our Figure 7, which leads to the misunderstanding of the reviewer. In the revised manuscript, we update texts in lines 188-189: "South of the 50 m isobaths, the density stratification hardly exists in winter months (Dec. Jan. Feb.), except for the Dutch coast and the German Bight. The potential energy anomaly increases since April and reaches …". In Line 193: "The intensity of the summer stratification varies."*

L215-216: this could be due to the use of in-situ temperature in the computation of water density, if this was the case.

*We use potential temperature in the computation of water density.  This is clarified in the revised manuscript.*

L227: How was the 50Jm-3 threshold chosen for the identification of stratified column?

*This criteria value implies the amount of energy required to instantaneously homogenise a 60 m depth water column with a surface-to-bottom density difference of 1 kg/m$^3$, where 60 meters is approximately the average water depth of the North Sea (excluding the Norwegian Channel). This is clarified in the revised manuscript line 186: "The water column is considered to be straitified when ɸ exceeds 50 J m$^{-3}$, which implies the amount of energy required to instantaneously homogenise a 60 m depth water column with a surface-to-bottom density difference of 1 kg/m$^3$, where 60 meters is approximately the average water depth of the North Sea (excluding the Norwegian Channel). ".*

L228 and L245-246: I do see the correlation between stratification and occurrence of heatwaves, but my impression is that the causality here could be somewhat overrated. Of course stratification hampers the transfer of heat along the water column, and therefore it is likely to be intensified in the occurrence of a MHW, but I would say that the interesting point here is rather the role of winter-spring "preconditioning" and the quantification of the processes driving the redistribution of heat through the water column.

*We thank the reviewer for mentioning the role of winter cold spell in the summer stratificaiton and added in the revised manuscript line 239: "It is worth noting that not only the MHW causes density stratification in the water column, but on the other hand the stratification hampers the transfer of heat along the water column and intensifies the occurrence of a MHW." and in line 251: "It is also important to stress here the role of winter-spring cold spell, which reduces the SST (see Figure 4), thus*

*increases vertical instability of the water column and enhances vertical mixing. This causes a colder sea water than the average before the summer termperature rising, especially in the lower water layers via the redistribution of the heat through the water column. Hence, a more persistent thermal stratification is likely to be obtained in the southern North Sea."*

L246-247: Is this a standard criterion?

*We set the value only for this study based on our data as shown in Figure 5. To clarify this we revised text in Line 246: "… than the warming period averaged increase rate of the multiyear mean (see Figure 5, upper panel)"*

L319-320: Again, this could be related to how stratification and water density were defined. Please discuss.

In the revised text, it is clarified now after the potential energy anomaly (PEA) and water density are defined.

TECHNICAL COMMENTS

Abstract: The time reference of the study is not clear from the abstract, I would suggest to add a few words about this.

*Thank you for the suggestion. In the Abstract, the modified text reads: "The role of heatwave events responsible for the occurrence and persistence of thermal stratification was analysed through the simulation of the North Sea water temperature from 2011 to 2018 using a fully coupled hydrodynamic and wave model within the framework of the Geesthacht Coupled cOAstal model SysTem (GCOAST)."*

L26: is this really "fastest" or just "faster"?

*We revised this sentence as: "The North Sea is predicted to be warming twice faster than the global levels and …"*

L237: Is a "where" missing here?

*We fix the sentence as: "In the region, the water depth was greater than 50 m …"*

---

## Author Comment (AC2)

**Reply to the comments of reviewer 2 on the NHESS submission:**

**'The role of heat wave events on the occurrence and persistence of thermal stratification in the southern North Sea'**

The manuscript aims to investigate the contribution of marine heatwaves (MHW) to thermal stratification in the North Sea by using ocean-wave coupled numerical experiments performed during a 8-year period. Comparing with observations (in-situ and remote-sensing), they observed large stratification events in the summers of 2014 and 2018, when large surface-to-bottom differences in water temperature co-occurred with high air temperatures. They argue that in absence of turbulent mixing high air temperatures lead to high SST and intensified stratification, triggered also by the long memory of seawater compared to air to low temperatures in the cooling season (e.g. January-April).

Thermal stratification and MHWs show a well-defined dependency in areas, along the Danish coast, which were not well characterized in the literature as one of the density stratification regimes in the North Sea. The vertical structure of stratification (Ri) shows the water depth as the factor modulating the sensitivity of the stratification to the summer heatwave (max values observed in the middle of the water column).

The work is well developed and well placed within the framework of the scientific open issues about the effect of extreme climate events. The authors use robust analysis and provide an exhaustive physical interpretation of all the results obtained, which address the goals of the study. The publication is recommended.

Here follow some suggestions, which could help improve certain parts of the manuscript.

*We thank the Editor and the reviewer for their careful reading of the manuscript and for many constructive comments and suggestions, which were useful to improve the manuscript. Below, we present a point-to-point reply to all comments. The original text of the reviewers is in black and our reply is in italic blue.*

General Comments

1. The sensitivity of ocean-wave coupling is addressed in supplementary materials (SM), but either a dedicated Appendix on the wave-induced processes included in the coupling, or an additional Figure in SM on the sensitivity to the wave-induced turbulent mixing, could help the reader to fully understand the results (e.g. Lines 270-275).

*We appreciate the reviewer for this suggestion and add added texts in discussion (line 275) to show the mixed layer depth in the southern North Sea during summer months with/without wave-induced processes coupling: "Figure 12 shows the modelled seasonal mean mixed layer depth (MLD) of the*

[Figure]

Figure 12. Upper panels: The seasonal mean mixed layer depth (MLD) of the coupled NEMO-WAM model run. Lower panels: The relative changes of MLD when comparing the coupled run with the stand-alone NEMO (uncoupled) run. Dotted lines in the lower panels indicate the 0 % contour.

*coupled NEMO-WAM model run in 2018 and the relative changes when comparing the fully coupled run with the uncoupled run, i.e.,* $(\mathrm{MLD_{coupled}} - \mathrm{MLD_{uncoupled}})/\mathrm{MLD_{coupled}} \times 100\%$**.**

*Following the annual cycle of water temperature, MLD in the southern North Sea decreases from winter to summer and develops again in autumn. The MLD changes quicker in deeper regions than in shallow coastal areas. There is almost no seasonal cycle along with the German Bight coast. The large difference in MLD between the coupled and uncoupled run is found from spring to autumn, when stratification develops with the changing air temperature. In summer, the MLD of the coupled run is approximately 20~ 40% larger than that of the uncoupled run in the southern North Sea, whereas it is approximately 20% smaller north to the 54ᵒN. In autumn, the stratification disappears in the southern North Sea, where the MLD differences drop to -10~10%. In the north, where water depth is larger than 50 m, the MLD of the coupled run is approximately 20% larger than the MLD of the uncoupled run. The wave-induced processes change also the heat fluxes at the water surface (Figure 13). The net heat flux is overall positive ( from air to the sea) in spring and summer and from sea to the air in winter and autumn  The importance of wave forcing for ocean predictions in the North Sea has been demonstrated by Staneva et al., 2017. The wave-induced processes impact the distribution of the heat*

[Figure]

Figure 13. The same as Figure 12, but for the surface heat fluxes. In the upper panels, positive values denote net fluxes from the air the sea.

*fluxes. In summer, in the southern North Sea, the fluxes increase by about 20-40% while along the coastal, well-mixed area of the German Bight is the opposite and a reduction of 20 -50% is observed.*

2. The quality of the graphics and Figure captions could be improved (look at specific comments).

*We thank the reviewer for this suggestion and corrected the figures accordingly.*

Specific Comments

Introduction

Line 17: the reference IPCC, 2012 should be updated to AR6

*The reference is updated.*

Line 26: "In the North Sea …" - rephrase.

*Now it is rephrased as "The North Sea is predicted to be warming …" and shifted to line 33-35.*

Line 55: "who showed…" – rephrase.

*The revised sentence reads: "Recently, Klonaris et al. (2021) developed a state-of-the-art three dimensional hydrodynamic model based on ROMS and showed accurately reproduced thermohaline variations in the southern North Sea.".*

Line 56 "Considering the interface …" – rephrase.

*The revised sentence reads: "Models investigate thermodynamic air-ocean interface processes by coupling the interactions between the air-sea system (Ho-Hagemann et al., 2017; Stathopoulos et al., 2020)."*

Section 2

Line 99: Check the reference GEMORAR, 2019

*Now it is corrected to GEOMAR, 2019*

Line 102: omit "for the location".

*Corrected.*

Section 3

Line 158-159: rephrase. Something like: "The results were further compared with satellite data. Figure 4 shows an annual variation in the modelled and remote sensed SST at the NSB III platform location in 2018.

*We thank the reviewer for the suggestion and we updated the sentence accordingly in the revision.*

Line 194: multi-year mean.

*Corrected.*

Section 4

Line 232: "….the number of days when the water …"

*Corrected.*

Line 265: "…down to the bottom": it is hard to distinguish max Ri values at the bottom, and the statement is in contrast with the sentence at Lines 267-268. Consider rephrasing.

*This sentence is revised in the new manuscript as: "Moreover, at the Dogger Bank and NSB III, the maximum $R_i$ shifted from the upper water layers to the bottom during the early stage of the warming season."*

Line 277-295: It could be shorter and move to the conclusions.

*We agree with the reviewer to shorten this paragraph. However, we decided to keep the text in this section because these texts discussed the potential hydrodynamic response of the southern North Sea to the occurrence of extreme MHW events rather than the main finding of this work. The new texts now read:*

*"Apart from circulation-wave coupling, stratification itself also influenced circulation. Baroclinic circulation, a result of the balance between baroclinic forcing and friction, is more pronounced with stronger current velocities (larger vertical shear) in warmer climate conditions when the turbulent eddy viscosity is eliminated by the increased stratification. Huang et al. (1999) showed an enhancement of water transport from a weakly stratified situation in spring to a strongly stratified situation in summer in the Bohai Sea. In the North Sea, relations between the cycle of thermal fronts and baroclinic circulation were analyzed at different ranges of timescales (Lwiza et al., 1991; Luyten et al., 1999). Decreased vertical turbulent mixing reduces the magnitude and horizontal shear of the baroclinic currents. This was found to be important feedback from turbulent mixing to the frontal temperature gradients and baroclinic circulation (Luyten et al., 2003). These mechanisms would be further pronounced in case extreme events occur. By analyzing 40-year data, Schrum et al. (2003) identified unusual stratification due to wind and volume transport in the 1990s. Recently, Chen et al. (2021) investigated the heat budget of the North Sea. They demonstrated the modification of SST affected advective water and heat transport in the North Sea. Comparatively, extreme events, i.e., summer heatwaves, had more intensive influences on SST but within a relatively shorter timescale. Exploring the reaction of the regional water circulation to extreme events in the southern North Sea is an interesting topic and deserves further study."*

Comments about Figures

Figure 2:

The bottom panel should be either described in the manuscript or removed.

*We add the following text in Line 149: "The differences in water temperature between the model and the measurements are in the range of ±2◦C. The model error is smaller during the winter months than in the summer months"*

Caption

"Bottom panel: the differences …".

*The sentence is revised as: "The water temperature differences between the model and the in-situ measurements at the surface and the bottom."*

Figure 3

The text in the caption should be improved.

*The updated caption reads: "The in-situ measured and the model simulated water temperatures (in◦C) in July. At the Dogger Bank, measurements are obtained by the CTD profiler during the Poseidon campaign and at the MARNET stations NSB III and FINO-1, temperatures are measured at fixed water layers."*

Figure 4

The text in the caption should be improved.

*The updated caption reads: "Annual variation of SST (in◦C) at the NSB III platform in 2018. Apart from the satellite measurement (yellow curve) and the model simulation (purple curve), the 10th- and 90th-percentile SSTs from the multiyear statistic are shown as well. Based on the 10th/90th-percentile, one cold-spell and two MHWs are detected.".*

Figure 6

Check the unit (see palette). The text in the caption should be improved describing clearly what is displayed on the background. It is hard to distinguish between dashed and solid black lines.

Change to J m$^{-3}$ in log$_{10}$ scale

*Thank you. We have revised the figure following your suggestions.*

Figure 9

What about the solid line? See relevant comment for Figure 6.

*We add the following sentence: "The thin solid line indicates the location of 50 m depth. "*

Figure 10

See previous comments.

*We revised the caption accordingly.*